# Deep Learning Analysis for Predicting Tumor Spread through Air Space in Early-Stage Lung Adenocarcinoma Pathology Images

**DOI:** 10.3390/cancers16112132

**Published:** 2024-06-03

**Authors:** De-Xiang Ou, Chao-Wen Lu, Li-Wei Chen, Wen-Yao Lee, Hsiang-Wei Hu, Jen-Hao Chuang, Mong-Wei Lin, Kuan-Yu Chen, Ling-Ying Chiu, Jin-Shing Chen, Chung-Ming Chen, Min-Shu Hsieh

**Affiliations:** 1Department of Biomedical Engineering, College of Medicine and College of Engineering, National Taiwan University, Taipei 10617, Taiwan; d10528013@ntu.edu.tw (D.-X.O.); f04548034@ntu.edu.tw (L.-W.C.); r12528052@ntu.edu.tw (K.-Y.C.); 2Division of Thoracic Surgery, Department of Surgery, National Taiwan University Hospital and National Taiwan University College of Medicine, Taipei 100, Taiwan; i2363160@gmail.com (C.-W.L.); renhao912@hotmail.com (J.-H.C.);; 3Graduate Institute of Pathology, National Taiwan University College of Medicine, Taipei 100, Taiwan; 4Division of Thoracic Surgery, Department of Surgery, Fu Jen Catholic University Hospital, No. 69, Guizi Road, Taishan District, New Taipei City 24352, Taiwan; yao120903@gmail.com; 5Department of Pathology, National Taiwan University Hospital and National Taiwan University College of Medicine, Taipei 100, Taiwan; karasu1114@gmail.com; 6Institute of Medicine, Chung Shan Medical University, Taichung 40201, Taiwan; f98546021@ntu.edu.tw

**Keywords:** deep learning, spread through air space, lung adenocarcinoma, digital histology, whole slide image, pathology

## Abstract

**Simple Summary:**

This study included 227 patients, among whom 27.7% (63/227) were diagnosed with tumors spread through air spaces (STASs), which have been shown to be associated with shorter recurrence-free survival and poor prognosis. A prediction model was developed to forecast tumor STAS in early-stage lung adenocarcinoma pathology images. The radiomics prediction model demonstrated good performance, with an AUC value of 0.83. This prediction model can assist pathologists in the diagnostic processes of clinical practice.

**Abstract:**

The presence of spread through air spaces (STASs) in early-stage lung adenocarcinoma is a significant prognostic factor associated with disease recurrence and poor outcomes. Although current STAS detection methods rely on pathological examinations, the advent of artificial intelligence (AI) offers opportunities for automated histopathological image analysis. This study developed a deep learning (DL) model for STAS prediction and investigated the correlation between the prediction results and patient outcomes. To develop the DL-based STAS prediction model, 1053 digital pathology whole-slide images (WSIs) from the competition dataset were enrolled in the training set, and 227 WSIs from the National Taiwan University Hospital were enrolled for external validation. A YOLOv5-based framework comprising preprocessing, candidate detection, false-positive reduction, and patient-based prediction was proposed for STAS prediction. The model achieved an area under the curve (AUC) of 0.83 in predicting STAS presence, with 72% accuracy, 81% sensitivity, and 63% specificity. Additionally, the DL model demonstrated a prognostic value in disease-free survival compared to that of pathological evaluation. These findings suggest that DL-based STAS prediction could serve as an adjunctive screening tool and facilitate clinical decision-making in patients with early-stage lung adenocarcinoma.

## 1. Introduction

The presence of spread through air spaces (STASs) in early-stage lung adenocarcinoma is a significant prognostic factor associated with disease recurrence and poor outcomes [1,2]. STAS has been reported to be a significant risk factor for recurrence in small-sized NSCLCs treated with limited resection [3,4]. Among patients with STAS-positive T1 lung adenocarcinoma, those treated with lobectomy have been shown to have better outcomes than those treated with sublobar resection [3]. Traditionally, STAS detection relies on pathological examinations by experienced pathologists. However, the advent of artificial intelligence (AI) and deep learning offers new opportunities for automated analysis of histopathological images.

Recent advancements in AI and deep learning have revolutionized medical image analysis, particularly in the detection, segmentation, and classification of tumor tissues in histological images [5,6,7,8,9,10,11,12,13,14,15,16,17]. Numerous studies have highlighted the efficacy of deep learning models in extracting critical information from routine pathological images, offering valuable clinical insights [18,19,20,21,22,23,24,25,26]. For instance, deep learning has been utilized for quantitative image analysis to forecast disease progression patterns, prognoses, and other clinical outcomes [27,28,29,30,31]. Despite these advancements, there remains a paucity of study specifically addressing AI-based STAS prediction using histopathological images.

Most existing STAS prediction methods rely on radiomic features derived from computed tomography (CT) imaging [32,33,34,35,36,37,38,39,40,41]. These methods, while useful, often face limitations due to the complexity of feature extraction and model intricacies, which can hinder their effectiveness in clinical settings.

In contrast, deep learning approaches have shown promise in capturing intricate features within images more effectively. By employing end-to-end training, these methods enhance predictive capabilities and provide more accurate prognostic information [27,28,29,30,31]. For example, recent studies have demonstrated the application of deep learning models to various medical imaging tasks, achieving high performance metrics. Elazab et al. used a combination of YOLOv5 and ResNet50 for brain tumor detection and classification [26]. Tsuneki et al. used the EfficientNetB1 model in their study on multi-organ adenocarcinoma classification [21]. These examples highlight the potential of deep learning in medical image analysis.

Given the complexity and variability of histopathological images, our study seeks to explore the potential of deep learning models for STAS prediction. We hypothesize that by analyzing complex patterns and relationships within the histopathological images, our model can provide a reliable prediction of STAS presence. This involves not just detecting the presence of certain cells or structures but understanding their distribution, morphology, and spatial relationships.

Although previous studies have involved artificial intelligence (AI)-based interpretation of pathological slide images [5,6,7,8,9,10,11,12,13,14,15,16,17], AI-based STAS prediction studies are still limited. This study aims to analyze the whole slide image of pathological slides from patients with early-stage lung adenocarcinoma using an existing AI model to ascertain the presence of STAS. By training deep learning models on a substantial dataset of pathology images encompassing cases with and without STAS, models can be trained to discern subtle variances and features associated with STAS. Subsequently, these models can be deployed to evaluate new pathological images and provide predictions regarding the presence or absence of STAS.

## 2. Materials and Methods

### 2.1. Study Population

We used a dataset containing 1053 cases from the “Lung Adenocarcinoma Pathological Slide Image Tumor Airway Spread Detection Competition I” (https://tbrain.trendmicro.com.tw/Competitions/Details/21, accessed on 11 April 2022) as the training set for model development. Each case included a cropped digital pathology whole slide image (WSI) in the Joint Photographic Experts Group (JPEG) format and a corresponding extensible markup language (XML)-formatted STAS annotation file. The image data were collected from The Cancer Genome Atlas Program (TCGA), and annotations were provided by the Institute of Biomedical Informatics at National Yang-Ming Chiao Tung University. The testing data were collected between January 2017 and December 2017 during lung resections performed on 630 patients at the National Taiwan University Hospital for lung cancer treatment; 399 patients were diagnosed with adenocarcinoma. Digital pathology WSI was used to capture pathological hematoxylin and eosin-stained permanent section slides from 227 patients diagnosed with stage I lung adenocarcinoma. Staging was performed according to the 8th edition of the American Joint Committee on Cancer (AJCC) criteria for lung cancer. After surgery, the patients underwent regular chest computed tomography scans at 6-month intervals for follow-up evaluations. This retrospective study was approved by the Research Ethics Committee of NTUH (protocol code 202207035RIND; date of approval: 14 July 2023), and the requirement for informed patient consent was waived.

### 2.2. Pathological Data Review

Hematoxylin and eosin-stained slides of 227 resected stage I lung adenocarcinomas were reviewed by two experienced thoracic pathologists (M. S. H. and H. W. H.). The histological features, including STAS, were evaluated based on the 2021 World Health Organization (WHO) classification of thoracic tumors [19]. STAS was defined as the spread of tumor cells beyond the main tumor edge into the air spaces, the presence of STAS tumor cells in at least one airspace beyond the tumor edge, and the presence of tumor cell nests in multiple airspaces. Artifacts were identified as randomly situated clusters of tumor cells at the edge of the tissue section, lack of continuous spread in airspaces, normal benign pneumocytes or bronchial cells, and linear strips of cells detached from alveolar walls [42]. Histological grading followed the 2021 WHO classification criteria [42,43], categorizing tumors into grade 1 (lepidic-predominant with <20% high-grade patterns), grade 2 (acinar or papillary-predominant with <20% high-grade patterns), and grade 3 (tumors with ≥20% high-grade patterns). Digital pathology images were scanned using the Hamamatsu NanoZoomer S360 Digital Slide Scanner (Shizuoka, Japan) at ×40 magnification.

### 2.3. Pathological Spread through Air Space (STAS) Prediction Model Development

The overall model development procedure consisted of (1) preprocessing of the testing data, (2) STAS candidate detection, (3) false-positive reduction, and (4) patient-based STAS prediction. In the preprocessing step, the digital pathology WSI data were magnified and cropped to the same size to normalize the attributes of the input image. Subsequently, the STAS candidate detection model was trained to detect the potential candidate area that could be a tumor cell spread into the airspace. To select promising candidates belonging to the STAS, a false-positive reduction model was trained to remove low-confidence candidates. The selected candidates were integrated to predict the patient’s STAS presence. A flowchart is shown in Figure 1.

#### 2.3.1. Image Preprocessing

The training image provided by the competition was zoomed in and cropped to 20× magnification and 1716 × 942 pixels from the WSI, respectively. The digital pathology test images obtained from the NTUH were the original WSI scanned using a Hamamatsu NanoZoomer S360 digital slide scanner (Shizuoka, Japan). Therefore, to normalize the attributes of the images across the two cohorts, an image preprocessing method is proposed in this study. Specifically, the original WSI of the testing set is first zoomed in at 20× magnification. Subsequently, the zoomed-in WSI is partitioned into several non-overlapping patches with an image size of 1716 × 942 pixels, matching the image attributes of the training set. Eventually, 24,995 patches are produced from 227 cases.

#### 2.3.2. Spead through Air Space (STAS) Candidate Detection

Subsequent to image preprocessing, the training data were applied to train a deep learning model for detecting the conditions that could potentially indicate a cell spreading into the air. Given that You Only Look Once version 5 (YOLOv5) is characterized by rapid inference speed and high accuracy in object detection, it is currently a widely accepted detection model for object detection [44,45,46,47,48]. Thus, this study applied YOLOv5 for preliminary detection of STAS candidates using digital pathology images. Specifically, the digital pathology images (1716 × 942 pixels) are forwarded to the trained YOLOv5 to predict the STAS candidate bounding box, which is represented by the upper-left coordinates (x, y), width, and height. In the YOLOv5 training process, the original YOLOv5 loss function [44,45,46,47,48] is applied to guide the gradient update. A stochastic gradient descent optimizer was used at an initial learning rate of 0.01. Furthermore, the model training was stopped early when no further improvement in the validation loss was detected within 100 continuous epochs. Within the epoch limit, the model with the lowest validation loss was saved as the best model for cross-validation. The other training parameters were set as follows: batch size, 16; L1 regularization penalty term, 1 (fully connected layer). Our model was trained using Python (version 3.8), PyTorch (version 2.0.0), and PyTorch-Cuda (version 11.7) on an Ubuntu server with two Quadro RTX 3090 ti (NVIDIA Corporation, Santa Clara, CA, USA) graphic processing units.

#### 2.3.3. False-Positive Reduction

After the YOLOv5 model detected candidates from digital pathology images, low-confidence candidates were further excluded to reduce false positives. To achieve a false-positive reduction, a ResNet-18 [49] model was applied to predict low- or high-confidence candidates. Specifically, for model training, the detected candidate area was first resized to 128 × 128 pixels using bilinear interpolation. The resized candidates and their corresponding pathological ground truths were then applied to train ResNet-18 to predict whether the candidates had high STAS confidence or not. During the testing process, the same process was applied to predict the STAS confidence for each candidate. Once the confidence was predicted, this study applied a cut-off value of 0.5 to classify the strong-confidence candidates (≥0.5) from low-confidence candidates (<0.5). In the training process of ResNet-18, a loss function called CrossEntropyLoss was used to guide the gradient updates. The Adam optimizer was utilized with an initial learning rate of 0.001, and the model with the lowest validation loss was saved as the best model for cross-validation. The other training parameters were set as follows: batch size, 8; epochs, 30. The model was trained on the same software environment and Ubuntu server as YOLOv5.

#### 2.3.4. Patient-Based Spread through Air Space (STAS) Prediction

Once the candidates were selected by false-positive reduction, the low-confidence candidates were excluded; however, several strong-confidence candidates might still be included in a digital pathology image. Therefore, an integration method is required to aggregate these candidate results to finally identify the presence of STAS in a patient (hence termed patient-based STAS prediction). In this study, we propose an averaging method to aggregate the results of the candidates. Specifically, the confidence values for all high-confidence candidates within a digital pathology image are averaged into a value indicating patient-based confidence. The average confidence is then applied as the final predictor for patient-based STAS prediction. 

### 2.4. Correlation Analysis between Histological Grades and Model Prediction

Given the strong correlation between STAS and high histological grades [50,51,52,53,54], this study also investigated whether predicted confidence correlates with histological grades. In other words, we were interested in determining whether the higher the patient’s histological grade, the more candidates would be detected with strong confidence. To explore this correlation, patients were divided into three histological grades based on the latest grading system [40]. The distribution of low- and high-confidence candidates was calculated for each histological grade. Statistical methods were then applied to assess any significant differences in candidate distribution across the histological grades.

### 2.5. Statistical Analyses

Patient characteristics, pathological features, and peri-operative outcomes were analyzed using descriptive statistics. Categorical variables were presented as counts and percentages, while continuous variables were expressed as mean ± standard deviation. Kaplan–Meier survival curves were used to evaluate disease-free and overall survival. Two-sample t-tests compared the average confidence levels between patients with and without STAS. The chi-square test was employed to compare the distribution of low- and high-confidence candidates across histological grades. Statistical analyses were performed using IBM SPSS Statistics for Mac (version 25.0), with significance set at *p* < 0.05. 

## 3. Results

### 3.1. Patient Demographics and Clinicopathological Characteristics

A total of 227 patients were enrolled in this study. Table 1 presents an overview of patient demographics and clinicopathological features. The majority of our study cohort consisted of females (64.8%), and a significant proportion were non-smokers (83.3%). The mean age of the patients was 61.1 years. Among stage I patients, T1a lesions were predominant, accounting for 62.1% of the cases. The mean tumor size was 1.7 ± 1.0 cm. Sublobar resection was performed in 145 (63.9%) patients. Overall, 63 specimens (27.7%) exhibited STAS. In terms of tumor histological grading, 60 cases (26.4%) were grade 1, 126 cases (55.5%) were grade 2, and 32 cases (14.1%) were grade 3.

### 3.2. Peri-Operative Outcomes and Survival Analysis

No surgical mortalities occurred within 30 days. The average hospital stay after the operation was 4.1 ± 4.9 days. Two patients developed post-operative chylothorax. All the patients recovered with a low-fat diet and adequate chest tube drainage. One patient developed prolonged air leakage and atrial fibrillation after the surgery. The patient was discharged on post-operative day 18 with extended chest tube drainage and antiarrhythmic medications. The overall complication rate was 1.3%.

We conducted an analysis of disease-free survival and overall survival in patients with and without STAS, as interpreted by pathologists. Figure 2 illustrates the statistically significant decline in disease-free survival among patients with STAS (*p* = 0.01). However, there was no significant difference in overall survival between the two groups.

Figure 3 illustrates a remarkable reduction in disease-free survival among patients who underwent sublobar resection in the STAS-positive group compared to the STAS-negative group (*p* < 0.001). However, among patients who underwent lobectomy, there was no significant difference in the DFS between the STAS-positive and STAS-negative groups (*p* = 0.22).

### 3.3. Performance of Pathological Spread through Air Space (STAS) Prediction Model and Correlation Results between Different Histological Grades

#### Confidence and Histological Grades

In the testing set, STAS tumor prediction achieved an AUC of 0.83 (Figure 4A), an accuracy of 72% (163 of 227), and a specificity of 63% (111 of 163) while operating with a sensitivity of 81% (52 of 64; threshold of 0.30) (Table 2). Previous experimental results can be found in the Appendix A. Furthermore, as shown in Figure 5B, the average confidence level of patients with STAS was significantly higher than that of patients without STAS (*p* < 0.001). 

In the correlation analysis between confidence and histological grade (Table 3), there were 9498 candidates detected in grade 1, 35,296 in grade 2, and 11,698 in grade 3. For grade 1, 21% (1962/9498) of the candidates were strong-confidence candidates, 36% (12,842/35,296) were grade 2 candidates, and 51% (5934/11,698) were grade 3 candidates. Based on the statistical test, the results indicated that the proportion of strong-confidence candidates was significantly lower in grades 2 (*p* < 0.001) and 1 (*p* < 0.001) than in grade 3.

### 3.4. Survival Analysis Based on Artificail Intelligence (AI) Pathological Feature Analysis

The developed AI model was applied to our patient cohort. Figure 5 presents a comparison of disease-free survival and overall survival between the AI-predicted STAS-positive and STAS-negative groups. We observed a significant reduction in disease-free survival in the AI-predicted STAS-positive group (*p* = 0.005); however, there were no significant differences in overall survival (*p* = 0.921).

As shown in Figure 6, the AI-predicted STAS-positive group displayed a significantly worse disease-free survival when undergoing sublobar resection (*p* = 0.007). However, there were no significant differences in disease-free survival among this group when they underwent lobectomy (*p* = 0.22).

A comparison between the presence or absence of the STAS and the AI prediction results is shown in Figure 7. It can be observed that AI predictions for DFS in cases with and without STAS were similar to the pathological results. Moreover, DFS was worse in cases predicted by AI to have no STAS than in those predicted to have STAS, mirroring the pathological findings. Overall, there were no significant differences among the four groups.

## 4. Discussion

Previous studies have consistently reported that the presence of STAS in early-stage lung adenocarcinoma is associated with a higher risk of disease recurrence [55]. Our cohort also demonstrated similar results, as STAS was a significant factor associated with shorter disease-free survival (*p* = 0.01) but did not affect overall survival. Again, in agreement with previous research [3], our study showed that STAS was a significant factor for shorter disease-free survival only in patients undergoing sublobar resection and not in those who underwent lobectomy. 

Detecting STAS requires experienced pathologists who are well versed in the diagnostic criteria for true STAS and who meticulously examine tumor borders on all HE-stained slides. The development of an AI model as a screening tool for STAS detection would be intriguing. Currently, most research is focused on using AI to analyze patients’ computed tomography radiomic data and predict STAS presentation [35,56,57,58]. These studies reported prediction AUCs ranging from 0.75 to 0.84 and accuracies between 0.74 and 0.81 [2,32,33,50,51]. Our study could potentially be the first to employ an AI model that predicts STAS based on digital pathological slide images. Our AI model achieved an AUC of 0.83 in predicting STAS, with a sensitivity of 81% and a specificity of 63%. The accuracy was 72% (163 of 227), which is comparable to the prediction values reported in previous studies based on radiomics data.

The AI model in this study gave each case hundreds of STAS candidates with low and strong confidence. These candidates outnumbered the true STAS defined by strict pathological criteria, and most of these candidates did not fulfill the strict pathological criteria of STAS. Cases with stronger confidence candidates were more likely to be STAS-positive. By highlighting candidates with strong confidence in STAS on WSI, we believe that our AI model can serve as a screening tool for pathologists.

Importantly, the AI-predicted STAS-positive and STAS-negative groups showed remarkably similar results in disease-free survival and overall survival compared with the results of the pathologically defined STAS-positive and STAS-negative groups. The AI-predicted STAS-positive group had significantly shorter disease-free survival only in those who underwent sublobar resection. These findings suggest that our AI model is a valuable tool for predicting the prognosis of patients with stage I lung cancer. This can facilitate thoracic surgeons in tailoring post-operative treatment plans for patients.

This study had some limitations. Firstly, the AI model could only select candidates for STAS, and most of these candidates were not true STAS. More effort may be required to annotate each STAS nest in the testing set to improve the AI model. Additionally, training the model requires a large amount of data, and the available public datasets mainly provide images in JPEG format. Therefore, under these constraints, we chose to use JPEG images. However, using JPEG images may affect the results of image analysis, especially when applying deep learning algorithms, as JPEG compression reduces high spatial frequency information, which could be crucial for capturing subtle features.

Secondly, our AI model cannot identify tumor borders, and many candidates are located within the tumor. These candidates represent high-grade tumor patterns, such as micropapillary nests formed by tumor cells in glandular structures. Correlation analysis indicates that the proportion of strongly confident candidates is significantly higher in patients with grade 3 tumors than in those with grade 2 or 1 tumors. This result is consistent with previous reports, showing a strong correlation between histological grade 3 adenocarcinoma and STAS. This result is consistent with previous reports showing a strong correlation between histological grade 3 adenocarcinoma and STAS [18,28,29,30].

Thirdly, the testing cohort includes patients with stage I lung adenocarcinoma. Further studies are needed to determine whether this AI model has the same AUC for stage II–IV lung adenocarcinoma. The statistical power of this study is somewhat limited, possibly due to the small sample size. Furthermore, the effects of the different types of digital pathology WSI are unknown. 

In summary, the application of the developed AI model for the prediction of STAS to the research cohort yielded an AUC of 0.83 and an accuracy of 72%. The AI model also predicted the disease-free survival of patients with state-I lung adenocarcinoma. The results of disease-free and overall survival between STAS-positive and STAS-negative groups were similar between the pathologically defined and AI-predicted groups. This underscores the reliability of the model as a valuable tool for aiding clinicians in treatment decision-making. 

## 5. Conclusions

We employed deep learning techniques to create a predictive model to identify the presence of STAS on pathological slides. This model has the potential for practical clinical applications, aiding both thoracic surgeons and pathologists in making informed decisions regarding treatment selection.

## Figures and Tables

**Figure 1 cancers-16-02132-f001:**
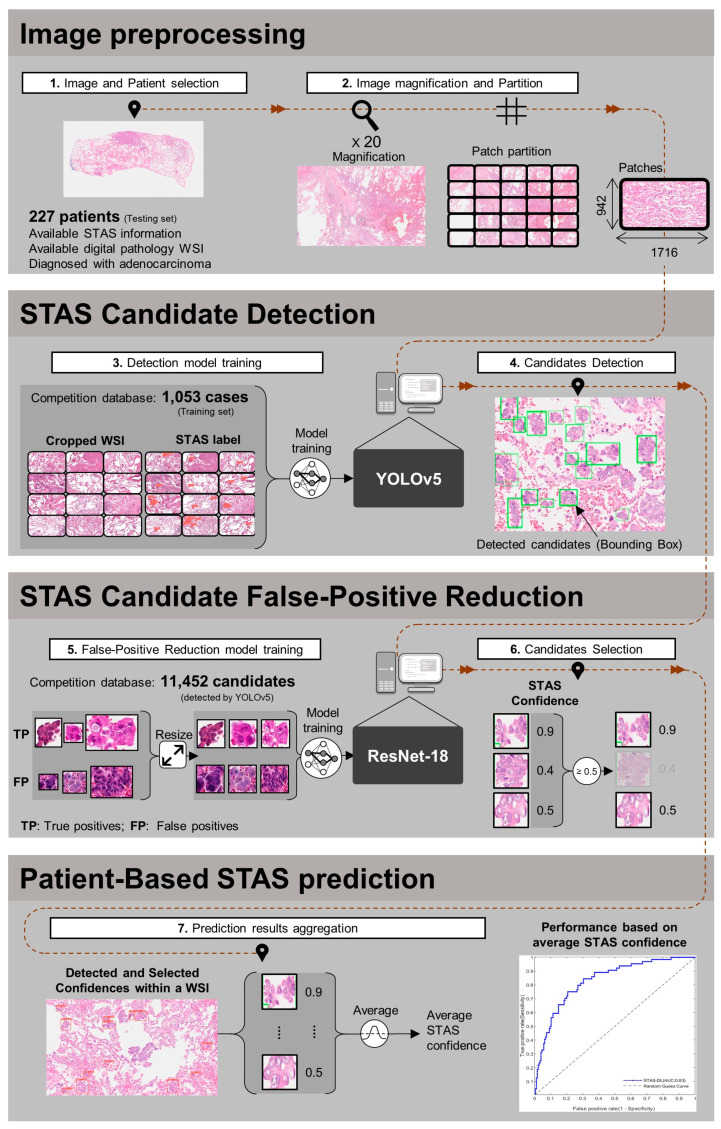
Framework for predicting tumor spread through air space (STASs) in early-stage lung adenocarcinoma. Summary of the analysis procedure: (1) Utilizing YOLOv5 for STAS detection training, with an image size of 1716 × 942, totaling 4212 images. (2) Cropping external test data to match the training data size, comprising 227 patients and 5800 images. (3) Further categorizing the detection results into STAS tumor positives/negatives using ResNet-18. (4) Evaluating the performance of the STAS tumor detection model with confidence > 0.5 as the criterion.

**Figure 2 cancers-16-02132-f002:**
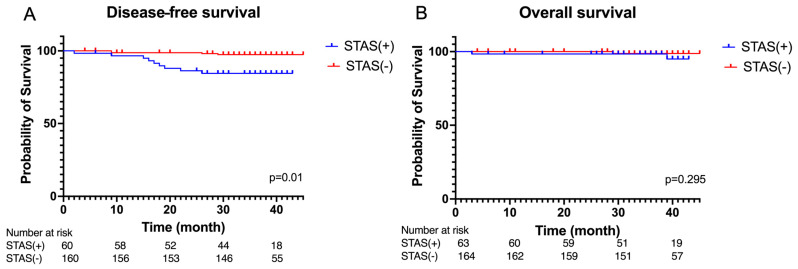
(**A**) Kaplan–Meier curves of disease-free survival and (**B**) overall survival of the STAS(+) group and STAS(−) group.

**Figure 3 cancers-16-02132-f003:**
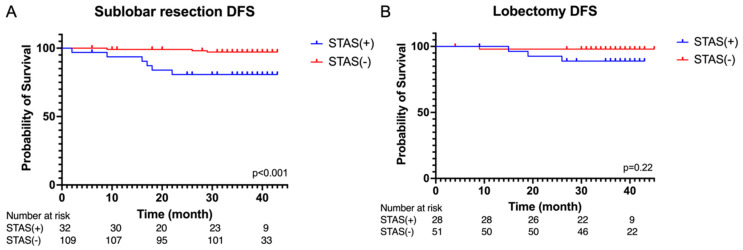
Kaplan–Meier curves of disease-free survival in the (**A**) sublobar resection and (**B**) lobectomy in the STAS(+) group and STAS(−) group.

**Figure 4 cancers-16-02132-f004:**
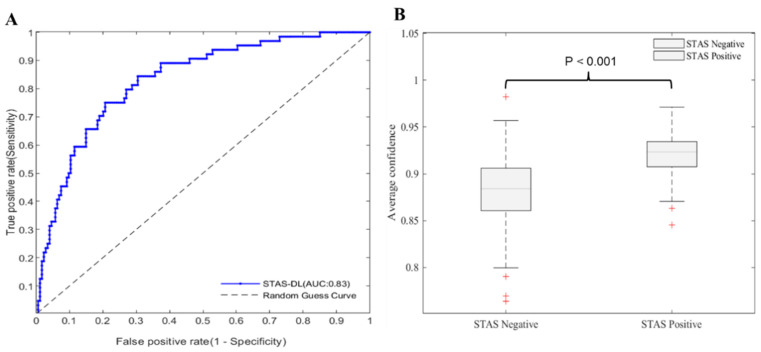
(**A**) Receiver operating characteristic curves (ROCs) for STAS prediction by the proposed method in the testing cohort (n = 227). (**B**) Distribution of predicted average confidence between patients with STAS positives and without STAS (STAS negatives). AUC, area under the ROC curve; STAS, spread through air space.

**Figure 5 cancers-16-02132-f005:**
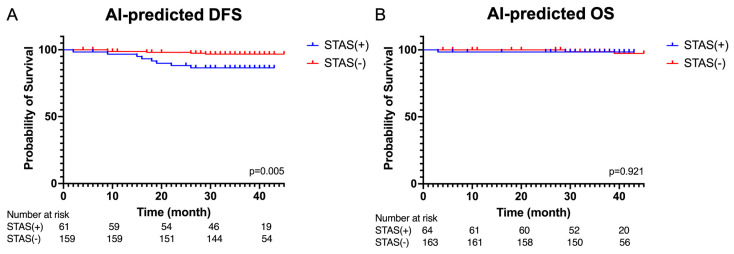
(**A**) Kaplan–Meier curves of disease-free survival and (**B**) overall survival of the AI-predicted STAS(+) group and STAS(−) group.

**Figure 6 cancers-16-02132-f006:**
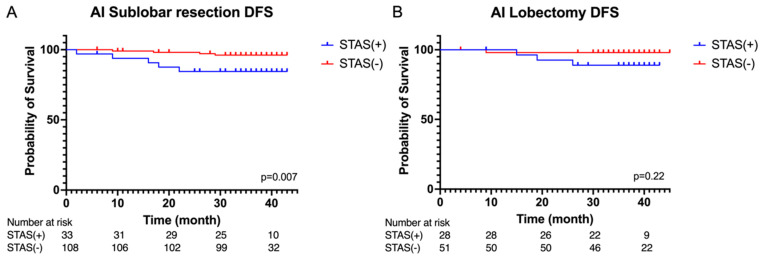
Kaplan–Meier curves of disease-free survival in the (**A**) sublobar resection and (**B**) lobectomy in the AI-predicted STAS(+) group and STAS(−) group.

**Figure 7 cancers-16-02132-f007:**
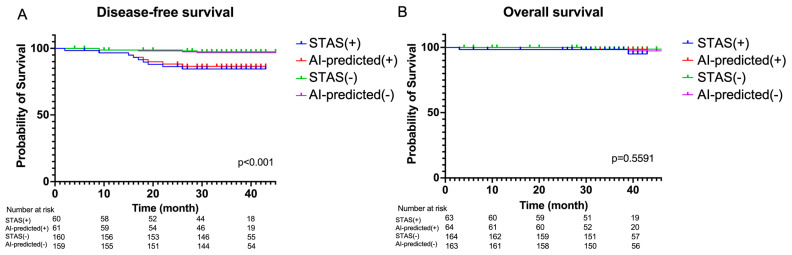
(**A**) Kaplan–Meier curves of disease-free survival and (**B**) overall survival between STAS present group, STAS absent group, AI-predicted STAS(+) group and STAS(−) group.

**Table 1 cancers-16-02132-t001:** Demographic and clinical features.

	N = 227
Age (year)	61.1 ± 10.5
Gender	
Female	147 (64.8%)
Male	80 (35.2%)
Smoking history	
Smoker	38 (16.7%)
Non-smoker	189 (83.3%)
Tumor size (cm)	1.7 ± 1.0
T stage	
T1a	141(62.1%)
T1b	31 (13.7%)
T2a	47 (20.7%)
Location	
RUL	82 (36.1%)
RML	19 (8.4%)
RLL	34 (15.0%)
LUL	58 (25.6%)
LLL	32 (14.1%)
Surgical procedure	
Lobectomy	82 (36.1%)
Sublobar resection	145 (63.9%)
Post-operative hospital stay (days)	4.1 ± 4.9
Complication	
Chylothorax	2 (0.9%)
Air leakage	1 (0.4%)
Atrial fibrillation	1 (0.4%)
STAS	
Present	63 (27.7%)
Absent	164 (72.3%)
Histological grading	
1	60 (26.4%)
2	126 (55.5%)
3	32 (14.1%)

Values are presented as n (%) or mean ± standard deviation. LUL, left upper lobe; LLL, left lower lobe; STAS, spread through the air space; RUL, right upper lobe; RML, right middle lobe; RLL, right lower lobe.

**Table 2 cancers-16-02132-t002:** Performance of STAS detection model.

Methods	Accuracy (%)	Sensitivity (%)	Specificity (%)	PPV (%)	NPV (%)	AUC (%)
Proposed model	72 (163/227)	81 (52/64)	68 (111/163)	50 (52/104)	90 (111/123)	83

The accuracy, sensitivity, specificity, PPV, NPV, and AUC are all presented as percentages. PPV: positive predictive value; NPV: negative predictive value; AUC: area under the receiver operating characteristic curve.

**Table 3 cancers-16-02132-t003:** Comparative analysis of tumor histological grades.

Histological Grades ^†^	Number of Detection Candidates	Number of Strong-Confidence Candidates * (%)	Number of Low-Confidence Candidates * (%)	*p*-Value **
Grade 1 (60/227)	9498	21 (1962/9498)	79 (7536/9498)	<0.001
Grade 2 (126/227)	35,296	36 (12,842/35,296)	64 (22,454/35,296)	<0.001
Grade 3 (32/227)	11,698	51 (5934/11,698)	49 (5764/11,698)	reference

The detection candidate confidence numbers are presented as percentages. * Data with confidence values greater than 0.5 are considered strong confidence candidates. Conversely, for data with confidence values less than 0.5, they are considered as low-confidence candidates. ** These statistics were calculated using the chi-square test and compared with grade 3. ^†^ There were nine patients without a completed pathological report; they were excluded from this correlation analysis.

## Data Availability

All data generated or analyzed during this study are included in this published article.

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
