# Peer review of "Deep Learning Analysis for Predicting Tumor Spread through Air Space in Early-Stage Lung Adenocarcinoma Pathology Images"

_cancers, 2024, doi:10.3390/cancers16112132_

Round 1

Reviewer 1 Report

Comments and Suggestions for Authors

The paper presents a very interesting approach to STAS prediction based on automated histopatological image analysis. While the results seem to be quite promising, a few issues must be addressed before the paper is accepted for publication.

1.      1.  am not a clinical expert, so there might be a very simple answer to this concern. It seems to me that if histopathology samples constitute the input then the question is not about STAS prediction but only automated detection.

2.      2. Comparison of data-based and analysis-based Kaplan-Meier curves for both DFS and OS proves that prediction quality is still not very satisfactory (as  are values of accuracy, specificity and sensitivity shown in Table 2). Therefore, the most important question is: about the superiority of the  proposed method over other, existing ones. There have been numerous papers on STAS published in recent years and the Authors should compare their method with others – actually that should be required in any research (unless the work is so pioneering that there is nothing to be compared to).

1.     3.  Another comment is related to the previous one – in the Introduction section the Authors should refer to other works that are focused on STAS prediction

Author Response

We acknowledge and value the time and effort you dedicated to reviewing the manuscript. Attached herewith are detailed responses addressing your feedback, along with the revised sections highlighted in the resubmitted documents. We are grateful for the valuable insights you have provided on our research.

Reviewer 2 Report

Comments and Suggestions for Authors

The manuscript aims to develop a deep learning (DL) model to predict the presence of spread through air spaces (STAS) in early-stage lung adenocarcinoma. This suggests that DL-based STAS prediction could be a valuable adjunctive tool for screening and aiding clinical decision-making in early-stage lung adenocarcinoma patients. The paper is well-written and straightforward, making it easy to follow. However, I provide a few comments that warrant further attention and clarification.

Major comments:

1. The literature review is inadequately covered. While the authors acknowledge the limited scope of AI-based STAS prediction studies in the paper, they fail to conduct a thorough review of pertinent literature (e.g., "DeepRePath: Identifying the Prognostic Features of Early-Stage Lung Adenocarcinoma Using Multi-Scale Pathology Images and Deep Convolutional Neural Networks"), which complicates the assessment of their study's progress and innovation. It is recommended that a concise overview of relevant papers be included, accompanied by an elucidation of the distinctive features of their developed model.

2.      There is no direct evidence provided regarding whether the inclusion of the False Positive Reduction step leads to improvement. It is recommended that the authors compare the effectiveness of STAS prediction with and without utilizing False Positive Reduction.

3.  Calculation speed is also a crucial consideration for screen tool development and its practical application in the field. It would be beneficial to evaluate the computational time required for processing a single case and provide a rough estimate to give readers a general idea.

Specific comments:

Line 65: repeat dots.

Line 363: template text should be removed.

Comments on the Quality of English Language

The quality of the English writing in this article is satisfactory. The content is well-structured and easy to follow, with clear explanations.

Author Response

(The authors gave the same response as above.)

Reviewer 3 Report

Comments and Suggestions for Authors

The paper proposes a method for tumor spread prediction through air space in lung adenocarcinoma images. The topic is quite new, difficult, and worth exploring. The experiments were carefully designed and carried out, therefore the presented results are reliable. However, one issue requires more attention and clarification. In the study, the authors used JPEG images. This is a lossy compression format that may affect the analysis results. JPEG compression does not usually affect the subjective impression of image quality because JPEG reduces some high spatial frequencies that are not perceived by the human visual cortex. However, this may impact deep learning algorithms, causing lower generalization. The obtained results, despite the large training data set, are not impressive (accuracy 0.7, specificity 0.68). Are such results sufficient for clinical application of the proposed system? Can using uncompressed images improve prediction results? Please address these issues in the revised version of this paper.

Author Response

(The authors gave the same response as above.)

Round 2

Reviewer 2 Report

Comments and Suggestions for Authors

The author has adequately addressed my questions. However, my suggestion is that the key points mentioned in the response were not incorporated into the manuscript. It is recommended to include these points directly in the manuscript or provide them as supplementary material for further clarification.

Reviewer 3 Report

Comments and Suggestions for Authors

Thank you for addressing my concerns. The paper is suitable for publication.